# Insights on the three-dimensional Lagrangian geometry of the Antarctic Polar Vortex

Jezabel Curbelo[1,2], Víctor José García-Garrido[1], Carlos Roberto Mechoso[3], Ana Maria Mancho[1], Stephen Wiggins[4], and Coumba Niang[1,5]

[1]Instituto de Ciencias Matemáticas, CSIC-UAM-UC3M-UCM. C/ Nicolás Cabrera 15, Campus de Cantoblanco UAM, 28049 Madrid, Spain.
[2]Departamento de Matemáticas, Facultad de Ciencias, Universidad Autonóma de Madrid, 28049 Madrid, Spain.
[3]Department of Atmospheric and Oceanic Sciences, University of California at Los Angeles, Los Angeles, California.
[4]School of Mathematics, University of Bristol. Bristol BS8 1TW, UK.
[5]Laboratoire de Physique de l'Atmosphere et de l'Ocean Simeon Fongang, Ecole Superieure Polytechnique, Universite Cheikh Anta Diop, 5085, Dakar-Fann, Senegal.

**Abstract.** In this paper we study the three-dimensional (3D) Lagrangian structures in the Stratospheric Polar Vortex (SPV) above Antarctica. We analyse and visualize these structures using the Lagrangian descriptor function $M$. The procedure for calculation with reanalysis data is explained. Benchmarks are computed and analysed that allow us to compare 2D and 3D aspects of Lagrangian transport. Dynamical systems concepts appropriate to 3D, such as normally hyperbolic invariant curves, are developed and applied. In order to illustrate our approach we select an interval of time in which the SPV is relatively undisturbed (August 1979) and an interval of rapid SPV changes (October 1979). Our results provide new insights on the Lagrangian structure of the vertical extension of the stratospheric polar vortex and its evolution. Our results also show complex Lagrangian patterns indicative of strong mixing processes in the upper troposphere and lower stratosphere. Finally, during the transition to summer in the late spring, we illustrate the vertical structure of two counterrotating vortices, one the polar and the other an emerging one, and the invariant separatrix that divides them.

## 1 Introduction

Over the past several decades the mathematical theory of dynamical systems has provided a fruitful framework to describe the transport and mixing proceses that take place in fluids and to understand the underlying flow structures associated with these phenomena. The seminal paper by Aref (1984) on chaotic advection sparked interest in this perspective, which is inspired by the work of Poincaré. For the understanding of particle dynamics, Poincaré sought a geometrical approach that was based on geometrical structures and their role in organizing all trajectories into regions corresponding to qualitatively different dynamical fates. These structures have been referred to as Lagrangian Coherent Structures (LCS) in the fluid mechanics community (Haller and Yuan, 2000; Shadden *et al.*, 2005).

Many studies of LCS in the atmosphere and in the ocean have been performed in a two-dimensional (2D) scenario. This is because in an appropriate range of space and time scales a Lagrangian property of the particles is approximately unchanged in time. Hence, the flow can be assumed to occur on surfaces on which that property is constant. For instance, stratospheric

flows in the time scale of stratospheric sudden warmings (∼10 days) are, to a first approximation, adiabatic and frictionless, and thus fluid particles and their trajectories are constrained to remain on surfaces of constant specific potential temperature (isentropic surfaces). Bowman (1993) and Joseph and Legras (2002) have examined transport processes across the Antarctic stratospheric polar vortex (SPV) on isentropic surfaces, which are quasi-horizontal in the atmosphere. Also, for oceanic flows

it is often assumed that fluid parcels remain on surfaces of constant density (isopycnals), which are quasi-horizontal. Mancho *et al.* (2006); d'Ovidio *et al.* (2009); Branicki *et al.* (2011) followed the isopycnal approach for oceanic applications in the Mediterranean Sea, Mendoza *et al.* (2014) for the Gulf of Mexico and Garcia-Garrido *et al.* (2015, 2016) for other ocean areas.

Geophysical flows, however, are not 2D. The study of transport processes in 3D flows brings into the discussion issues about the three-dimensional (3D) visualization of Lagrangian structures (see e.g. Wiggins (2010)). In idealized 3D time-

dependent flows Poincaré sections have been used to recognize significant Lagrangian structures (Cartwright *et al.*, 1996; Pouransari *et al.*, 2010; Moharana *et al.*, 2013; Rypina *et al.*, 2015). Invariant manifolds acting as transport barriers in 3D flows may have the structure of convoluted 2D surfaces embedded in a volume (Branicki and Wiggins, 2009). In oceanic contexts these surfaces have been identified by stitching together 2D Lagrangian structures at different layers (Branicki and Kirwan Jr., 2010) or connecting ridges computed from Finite Size Lyapunov exponent fields (Bettencourt *et al.*, 2014). In the field of

atmospheric sciences, these structures have similarly been obtained by connecting ridges computed from Finite Time Lyapunov Exponents (FTLE) (Rutherford and Dangelmayr, 2010; du Toit and Marsden, 2010; Lekien and Ross, 2010). More recently, also in atmospheric contexts, 3D Lagrangian information has been extracted by means of 2D slices of the full 3D FTLE field computed from 3D trajectories (Rutherford *et al.*, 2012).

The methodology used in this paper for visualizing 3D Lagrangian structures in the stratosphere is based on the Lagrangian

descriptor known as the function $M$ (Madrid and Mancho, 2009; Mendoza and Mancho, 2010; Mancho *et al.*, 2013). So far, in the stratosphere context, the function $M$ has been used to gain insight into key dynamical and transport processes in 2D settings (de la Cámara *et al.*, 2012, 2013; Smith and McDonald, 2014; Guha *et al.*, 2016; Manney and Lawrence, 2016; García-Garrido *et al.*, 2017). More recently, (Mancho *et al.*, 2013; Lopesino *et al.*, 2017) have applied the function $M$ to the visualization of structures in idealized 3D flows. Rempel *et al.* (2013) have applied the function $M$ to visualize coherent structures in full

3D direct numerical simulations of the compressible magnetohydrodynamic equations. The function $M$ has the advantage of highlighting simultaneously invariant manifolds by means of singular features and also tori-like coherent structures (see Mendoza and Mancho (2010); de la Cámara *et al.* (2012); Rempel *et al.* (2013); Mezic and Wiggins (1999); Lopesino *et al.* (2017)). In here we apply $M$ to produce a full 3D description from 3D flows above Antarctica during a period in the spring of 1979 in which the stratosphere was both rather stable (August) and subjected to rapid changes (October) (Yamazaki and

Mechoso, 1985). In this region, the later period selected for analysis comprises an interval when the winter circulation, which is characterized by a strong circumpolar westerly (cyclonic) flow known as the Stratospheric Polar Vortex (SPV), breaks down as the final warming to summer condition develops. Although final warmings in the southern hemisphere are broadly similar each year (Mechoso *et al.*, 1988), can be punctuated by periods of rapid changes. During the final warming of 1979, the transition from the winter to the summer circulation accelerated during mid-October, a period when perturbing waves were

very active and the vertical energy flux from the troposphere intensified (Yamazaki and Mechoso, 1985). Our aim is to describe

and visualize these phenomena from a full 3D perspective using reanalysis data. To our knowledge, this is the first time that the potential of $M$ to achieve this goal is explored.

The article is organized as follows. Section 2 describes the dynamical systems approach to the analysis of 3D Lagrangian structures. Section 3 describes the dataset used in this study, the calculation of $M$ in 3D, the data post-processing needed to this end, the computational procedures and other issues involved in this task. Section 4 discusses some benchmark calculations and their interpretation. Section 5 provides the results and findings of $M$ on the 3D Lagrangian structure of the polar vortex on August and mid-October 1979. Finally, section 6 includes a discussion and presents our conclusions.

## 2 The Dynamical Systems Approach to the Analysis of 3D Lagrangian Structures

The theory of dynamical systems provides an ideal framework for studying nonlinear transport and mixing processes in the atmosphere. The geometrical structures that vertebrate the Lagrangian skeleton act as material barriers that fluid particles cannot cross. A key element on the dynamical description are the presence of hyperbolic regions defined by rapid fluid contracting and expanding rates along directions that are respectively associated to the stable and unstable manifolds. In 2D flows, these manifolds are curves while in 3D settings other possibilities arise. We discuss some particularities for the system under study next.

If we assume that air parcels are passively advected by the flow, the dynamical system that governs the atmospheric flow is given by:

$$\dot{\mathbf{x}} = \mathbf{v}\left(\mathbf{x}\left(t\right),t\right) \;\; , \;\; \mathbf{x}\left(t_0\right) = \mathbf{x}_0 \; , \tag{1}$$

where $\mathbf{x}\left(t;\mathbf{x}_0\right)$ represents the trajectory of an air parcel that at time $t_0$ is at position $\mathbf{x}_0$, and $\mathbf{v}$ is the velocity field. For the geophysical context we are focusing on, the velocity components will be supplied by the ERA-Interim reanalysis dataset produced at the European Centre for Medium-Range Weather Forecast as explained in detail in the next section. As it will be verified in there, the magnitude of the vertical velocity component in the middle and upper stratosphere is very small so that vertical displacements of fluid parcels compared to the horizontal displacements are also small for the time scales of interest in this study ($\sim$days). These considerations motivate the discussion of a system with the particular structure given in Eq. (2) as it will support the interpretation of the findings described in Section 5 in this region of the atmosphere.

For two dimensional flows hyperbolic trajectories and their stable and unstable manifolds are the key kinematical features responsible for the geometrical template governing transport. However, in three dimensions there are new types of three dimensional structures that may form a geometrical template that governs transport. The ''weak three dimensionality'' of many geophysical flows, like the one considered in this paper, gives rise to a (normally) hyperbolic invariant curve (i.e. not a single trajectory) that has two dimensional stable and unstable manifolds embedded in the 3D space. In this case the stable and unstable manifolds of this invariant curve are codimension one in the flow and therefore provide barriers to transport. Moreover, since in this case the stable and unstable manifolds are both codimension one they can intersect to form lobes, resulting in a three dimensional version of lobe dynamics. We now describe the special form of the flow giving rise to this structure. The

form of the flow follows from Wiggins (1988) and was described in the context of fluid mechanics in Mezić and Wiggins (1994).

Here we examine a simple model three dimensional velocity field that captures the form of the velocity field given by the data set that we study and, therefore, allows us to describe this less familiar notion of a normally hyperbolic invariant curve in a simple setting. The velocity field has the following form:

$$
\begin{cases}
\dfrac{dx}{dt} = \dfrac{\partial H(x,y,z,t)}{\partial y} = v_x(x,y,z,t) \\[2mm]
\dfrac{dy}{dt} = -\dfrac{\partial H(x,y,z,t)}{\partial x} = v_y(x,y,z,t) \quad , \\[2mm]
\dfrac{dz}{dt} = 0
\end{cases}
\tag{2}
$$

where $H(x,y,x,t) = A(z)\sin(\pi y)\sin(\pi x)/\pi$ and $A(z) = 1 + \sin(\pi z/2)$. The system is defined in the domain $(x,y,z) \in [0,1] \times [-1,1] \times [-1,1]$. We refer to $x$ and $y$ as the horizontal coordinates and $z$ as the vertical coordinate. This velocity field is certainly ''weakly" three dimensional as there is no motion in the vertical direction yet the horizontal motion does depend on the height. This model contains the essence of the geometrical structures governing transport in the data set that we analyze.

More specifically, note that for each $z$ the system (2) has a hyperbolic fixed point at $(x=0, y=0)$, for which the linearized system is:

$$
\frac{dx}{dt} = A(z)\pi x, \tag{3}
$$
$$
\frac{dy}{dt} = -A(z)\pi y. \tag{4}
$$

The curve $(0,0,z)$ is clearly an invariant curve (in particular, it is a curve of fixed points). The linearized stability described by (4) quantifies linearized stability *normal* to the invariant curve $(0,0,z)$ and, hence, is the origin of the phrase *normal hyperbolicity*. For each $z$, the fixed point has a one dimensional stable manifold and a one dimensional unstable manifold. Therefore as a function of $z$ the curve $(0,0,z)$ has two dimensional stable and two dimensional unstable manifolds in three dimensions. Hence, these two dimensional invariant surfaces provide barriers to transport in the three dimensional flow (Wiggins, 1988; Mezić and Wiggins, 1994). The geometrical representation of the manifolds is shown in Figure 1a).

The 3D Lagrangian structure of Eq. (2) visualised in Figure 1b), is achieved by means of the function $M$, which is represented on slices intersecting the manifolds. Other slice choices are possible, but here we just show two simple possibilities on perpendicular planes which help to capture the essential features of the full 3D motion. The function $M$ is defined as follows:

$$
M(\mathbf{x}_0, t_0, \tau) = \int_{t_0-\tau}^{t_0+\tau} \|\mathbf{v}(\mathbf{x}(t; \mathbf{x}_0), t)\| \, dt , \tag{5}
$$

where $\mathbf{v}(\mathbf{x}, t)$ is the velocity field and $\|\cdot\|$ denotes the Euclidean norm. At a given time $t_0$, $M$ corresponds to the length of the trajectory traced by a fluid parcel starting at $\mathbf{x}_0 = \mathbf{x}(t_0)$ as it evolves forwards and backwards in time for a time interval $\tau$. For

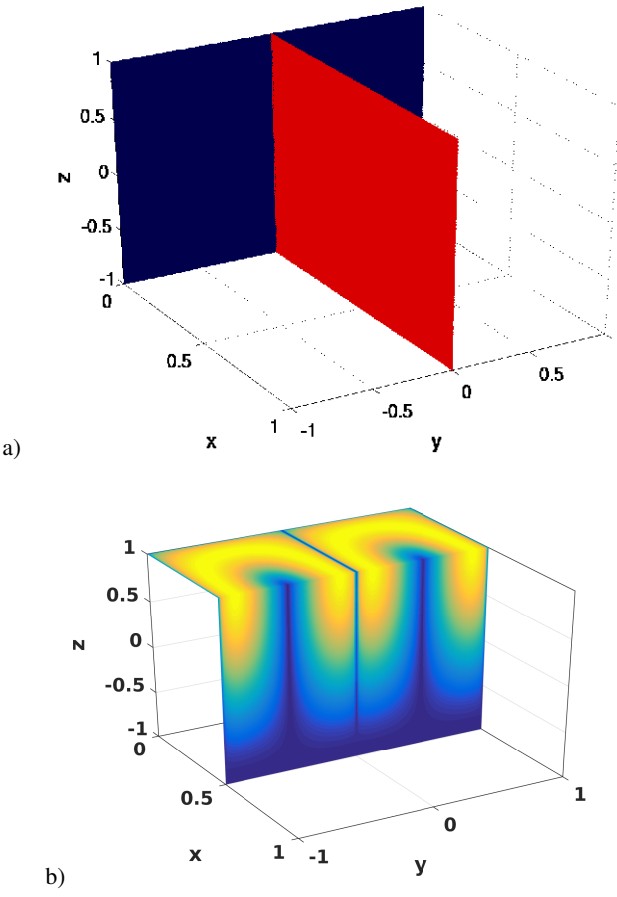

**Figure 1.** a) Stable (blue) and unstable (red) manifolds of the normally hyperbolic invariant curve in Eq. (2); b) representation of the Lagrangian structures at specific slices by means of the function $M$.

sufficiently large $\tau$ values the sharp changes that occur in narrow gaps in the scalar field provided by $M$, which we will refer to as singular features, highlight the stable and unstable manifolds and, at their crossings, hyperbolic trajectories as confirmed by Figure 1b). Recently, Lopesino *et al.* (2017) have established a rigorous mathematical foundation for specific LDs for a class of examples in continuous time dynamical systems.

5    Figure 1b) shows that the function $M$ has also the capability of revealing vortices present in the fluid. In particular for this example two counterrotating vortices which are vertically extended are visible. The yellowish colours highlight the part of vortices with highest speeds. Typically vortex or jet like structures (for periodic domains) are related to 2-tori in three dimensional flows. This is discussed in Mezić and Wiggins (1994); Wiggins (2010). This notion is related to fluid regions trapping fluid parcels in their interior and isolating them from the surrounding fluid as for instance is the case of the circulating

10    strong jet forming the SPV. There exist formal results linking contourlines of the time average of $M$ with tori-like invariant

sets. In this manner singular lines in 1b) highlight invariant manifolds and contourlines of converged averages of $M$ highlight invariant tori (see Lopesino *et al.* (2017)).

## 3 Dataset and computation of the function $M$

### 3.1 ERA-Interim Reanalysis dataset

We use the ERA-Interim Reanalysis dataset produced by European Centre for Medium-Range Weather Forecasts (ECMWF;Simmons *et al.* (2007)). The usefulness of this dataset for the Lagrangian study of atmospheric flows from the dynamical systems perspective is established by the results of several previous studies. For instance, de la Cámara *et al.* (2013) applied Lagrangian Descriptors (LDs) to study the structure of the SPV during the southern spring of 2005, to support the interpretation of several features found in the trajectories of superpressure balloons released from Antarctica by the VORCORE project (Rabier

*et al.*, 2010). ERA-Interim covers the period from 1979 to the present day (Dee *et al.*, 2011), and can be downloaded from http://apps.ecmwf.int/datasets/data/interim-full-daily/levtype=sfc/.

From the ERA-Interim dataset we extract the 3D wind velocity components, potential vorticity, surface pressure, and the geopotential field. In the version of the dataset that we selected for the present study, these physical variables are available four times daily (00:00 06:00 12:00 18:00 UTC) with a horizontal resolution of $0.75° \times 0.75°$ in longitude and latitude. The velocity

fields extracted correspond to the 60 hybrid-sigma levels of the model component of the reanalysis system (from the Earth's surface to the 0.1 hPa level) . We also take from the dataset potential vorticity at 15 levels of potential temperature (265, 275, 285, 300, 315, 330, 350, 370, 395, 430, 475, 530, 600, 700, 850; K), and geopotential at field pressure levels (1, 2, 3, 5, 7, 10, 20, 30, 50, 70, 100 to 250 by 25, 300 to 750 by 50, 775 to 1000 by 25; hPa).

### 3.2 Computation of the function $M$

The procedure to obtain the function $M$ in (lat, lon, height) coordinates from the data described in the previous section consists of the following steps.

Step 1. The data is downloaded from ERA-Interim in `.grib` format and on a monthly basis.

Step 2. The data files are converted from `.grib` to `.nc` format. This is done with the command `copy`, setting as arguments `-t ecmwf` to indicate that the data is from ERA-Interim and `-f nc` to specify that the output complies with NetCDF.

Step 3. The 3D velocity and 2D surface pressure data are concatenated to provide the input to the Climate Data Operator (CDO) software (available at https://code.zmaw.de/projects/cdo) using the CDO command `merge`.

Step 4. The data is converted from sigma levels to the height levels specified for the analysis, and required data are produced by interpolation, with the vertical velocity expressed in ms$^{-1}$. This is done using the CDO command `ml2hlx` applied to the NetCDF files. The resulting 3D velocity field has a horizontal resolution of $0.75° \times 0.75°$ in longitude and latitude, with

80 height levels ranging from 0 m to 47600 m at 600 m intervals. Each wind data variable is stored in separate files by day

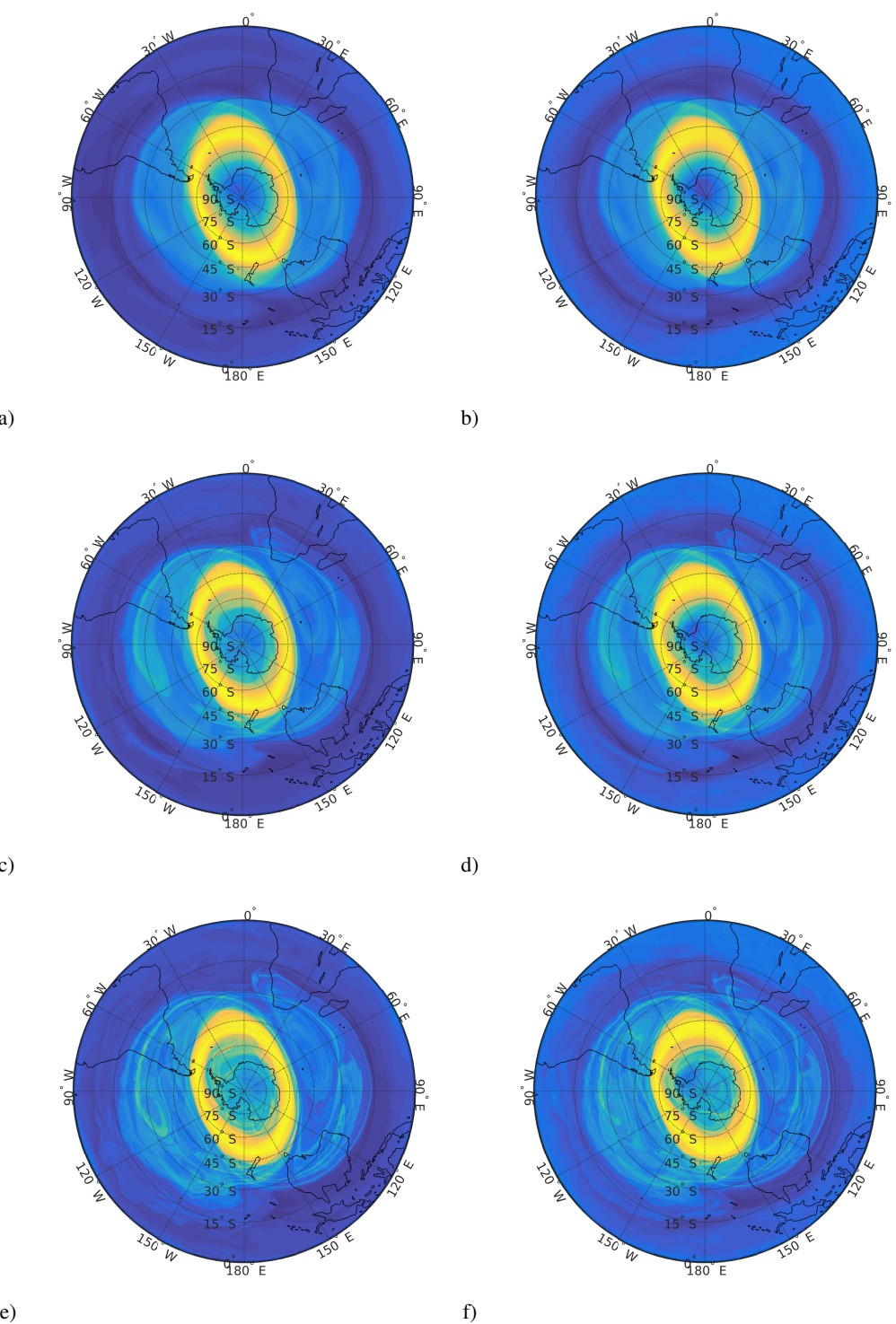

**Figure 2.** $M$ calculated on the 15th August at 1979 00:00:00 UTC: a) using a 2D approach where particle trajectories are constrained to the 850K potential temperature level and $\tau = 5$ days; b) $M$ computation for $\tau = 5$ days using a full 3D computation of trajectories and it is represented at a constant height level of $h = 31.3$ km (which corresponds approximately to the 850K isentropic surface); c) the same as a) with $\tau = 10$ days ; d) the same as b) with $\tau = 20$ days ; e) the same as a) with $\tau = 20$ days.

using the commands `selvar` and `selday`. This procedure avoids the handling very large data files when computing particle trajectories, which requires interpolation in time and space.

Step 5. To avoid issues at the pole in the calculation of trajectories from equations expressed in spherical coordinates, we write the velocities in cartesian coordinates from the velocity data available in spherical coordinates (see de la Cámara *et al.* (2012)); The velocity components in cartesian coordinates are given by,

$$
\begin{cases}
v_x = w \cos \lambda \cos \phi - u \sin \lambda - v \cos \lambda \sin \phi, \\
v_y = w \sin \lambda \cos \phi + u \cos \lambda - v \sin \lambda \sin \phi, \quad \cdot \\
v_z = w \sin \phi + v \cos \phi,
\end{cases}
\tag{6}
$$

where $u$, $v$, and $w$ are the zonal, meridional, and vertical velocity components, respectively, $\lambda$ is longitude, and $\phi$ is latitude.

Step 6. The trajectories are calculated in a cartesian coordinate system, which are obtained by solving,

$$
\begin{cases}
\dfrac{dx}{dt} = v_x(\lambda, \phi, h, t) \\[2mm]
\dfrac{dy}{dt} = v_y(\lambda, \phi, h, t) \quad , \\[2mm]
\dfrac{dz}{dt} = v_z(\lambda, \phi, h, t)
\end{cases}
\tag{7}
$$

To integrate the system (7) the data interpolation of the fields $v_x$, $v_y$, $v_z$ is carried out by interpolating $u$, $v$, $w$ in spherical coordinates, since the post-processed data files from ERA-Interim are expressed in this way. In order to do so, we have used the `griddedInterpolant` function provided by the MATLAB© software, which generates an object that can be saved into memory and evaluated at any point of interest in the trajectory at a later time. This allows for substantial saving in computational time at every integration step. Moreover, when applying this function to interpolate the dataset, we specify that a cubic interpolation is used in space and also in time. The system (7) is integrated using a Cash-Karp Runge-Kutta scheme (Press *et al.*, 1992) with a time step of 1 hour. The condition $w = 0$ is imposed at $h = 0$ in order to constrain trajectories in the vertical. Important computational savings are also achieved by integrating simultaneously the whole meshgrid of initial conditions by using a matrix formulation. Additionally, the computational time is reduced by running the MATLAB© software with options requiring the use of multiple cores.

Step 7. The function $M$ is obtained by approximating the integral in (5) by the sum of the lengths (in the euclidean space) of the segments linking the position of the integrated particle trajectory at two successive time steps. As $\tau$ increases a richer Lagrangian history is incorporated into $M$, and a more complex and detailed dynamical description is obtained.

We note that for the ERA interim data set $w$ is very small in the upper stratosphere at 31.3 km (see Table 1 for values during the period September-October 1979). Moreover, the mean value of $w$ at that level is 100 times less than the maximum value, with a very low mean dispersion. Table 1 also shows that in the upper troposphere levels at 10 km) the values of $w$ are comparatively not as small.

| $h$ | Max($u$) | Max($v$) | Max($w$) | $\overline{w}$ | $\sigma_w$ |
|---|---|---|---|---|---|
| 31.3 km | 132.49m/s | 100.07 m/s | 0.14m/s | 0.0021 | $6.2\ 10^{-4}$ |
| 10 km | 107.32m/s | 80.31 m/s | 1.37m/s | 0.049 | 0.23 |

**Table 1.** Comparison of the values taken by velocity components at different atmospheric levels.

## 4 Benchmarks

To benchmark the procedure described in the previous section, we compare the Lagrangian outputs obtained from the full 3D scenario at constant heights (spherical shells) with those obtained from the 2D scenario for potential temperature surfaces that are approximately at the same height. As expected from results in table 1, given that $w \sim 0$ in the upper stratosphere, the potential temperature surface and the spherical shell are expected to be almost identical. Figure 2 compares for different $\tau$ values, the evaluation of the function $M$ for a 2D integration in the 850K isentropic surface with that obtained on a spherical shell at 31.3 km height, which approximately corresponds to the 850K surface. For $\tau = 5, 10$ days the figure highlights well defined structures (de la Cámara *et al.*, 2012) that are very similar in both cases. These consist of a large circulating coherent vortex in yellow, representing high values of $M$ that are related to particles exhibiting large displacements and blueish zones corresponding to calmer regions. Lobes eroding the outer part of the vortex are clearly identified by colored filaments. Also crossings of contours of $M$ highlighting hyperbolic trajectories are noticed along longitudes 15$^o$W and 165 $^o$E.

Increasing the $\tau$ values to 20 or 30 days adds more Lagrangian detail to the figures, which thus contain long term transport issues reflected in very thin and long filamenteous structures. However the figures at constant potential temperature and constant height are still very similar and contain the gross structure already observed at smaller $\tau$. Differences are restricted to those filamenteous structures which are difficult to follow and to compare. Given that long term transport issues are difficult to interpretate and that we do not require them for describing the phenomena of our interest which occur in time intervals varying from hours to 10 days, we fix in this range the selected $\tau$ values for this article.

Comparisons of transport between both approaches in the upper troposphere are rather different, as figure 3 confirms. In this case table 1 shows significant non zero vertical velocities, and thus surfaces of constant potential vorticity are expected not to be close to simple spherical shells. The first row shows the 2D calculations for a 2D integration in the 330K isentropic surface for a period of $\tau = 5, 10$ days and the second row a 3D output at the corresponding height (10km) for the same integration intervals. Clear differences are observed in the images that are visible even at the shorter integration period of $\tau = 5$ days. This confirms that in this case the potential temperature surface is not close to the spherical shell for which the 3D calculations are shown. Results discussed in the next section will show further evidence of the 2D-3D motions transitions.

Figure 4 contrasts the more classical description of events in the middle-upper stratosphere with the one provided by our Lagrangian descriptor. The left hand column of Figure 4 shows the geopotential height at 10mb. This field is particularly useful because it is equivalent to the velocity streamfunction at the corresponding presure level in the extratropics (see García-

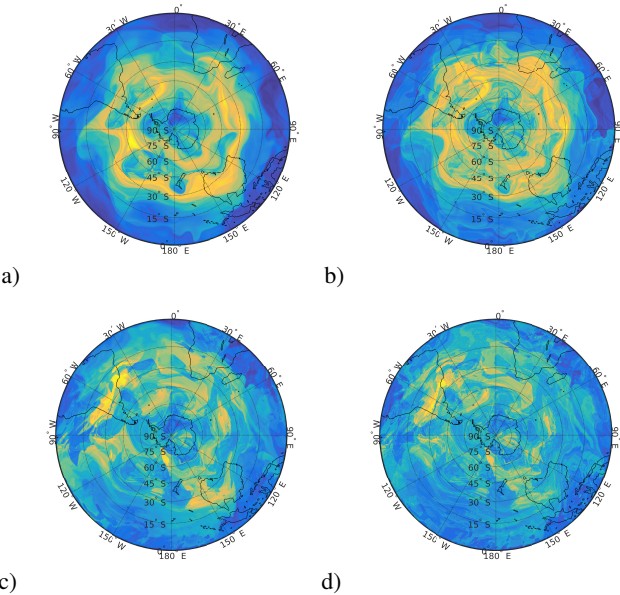

**Figure 3.** $M$ calculated on the 15th August at 1979 00:00:00 UTC: a) using a 2D approach where particle trajectories are constrained to the 330K potential temperature level and $\tau = 5$ days; b) the same with $\tau = 10$ days; c) $M$ calculated for $\tau = 5$ days using a 3D computation of trajectories and represented at a constant height level of $h = 10$ km (which corresponds approximately to the 330K isentropic surface); d) the same with $\tau = 10$ days.

Garrido *et al.* (2017)). This field provides a purely Eulerian description of the flow, however in the time dependent case this point of view is limited as it does not address issues regarding the fate of particle trajectories. The middle column shows the picture provided by the potential vorticity at the 850K equivalent temperature (isentropic) surface. This field is conserved along particle trajectories, thus the image is Lagrangian in the sense that we observe the field of a purely advected quantity. The right hand column shows the function $M$ at the surface $z = 31.3$ km obtained from particle trajectories in full 3D calculations. Similarly to the second column the displayed information is Lagrangian, but here we obtain more fundamental information in this regard. This figure provides feedback for characterising the time evolution of any purely advected scalar field, while the previous one displays just the realisation of one particular initial data. More specifically, the third column highlights the position and evolution of two hyperbolic points in the outer part of the vortex, as well as the vortex itself. As discussed by García-Garrido *et al.* (2017) hyperbolic points are responsible for filamentation processes. Whether or not these filaments are eventually observed depends on the distribution of the scalar field. For instance if the scalar field is completely uniform in the whole domain then its time evolution will show nothing about the features highlighted by $M$. How the features of the $M$ field are visible in a scalar field depends on how is the initial distribution of the advected field, with respect to the features of $M$. Figure 5 illustrates these facts in a very simple example. Figure 5 a) highlights the Lagrangian skeleton as obtained for the stationary cat eyes. The hyperbolic fixed point at the origin and its stable and unstable manifolds are clearly visible. Also elliptic fixed points at positions $(-\pi, 0)$ and $(\pi, 0)$ are visible. Figure b) shows a set of initial scalar fields and c) shows the

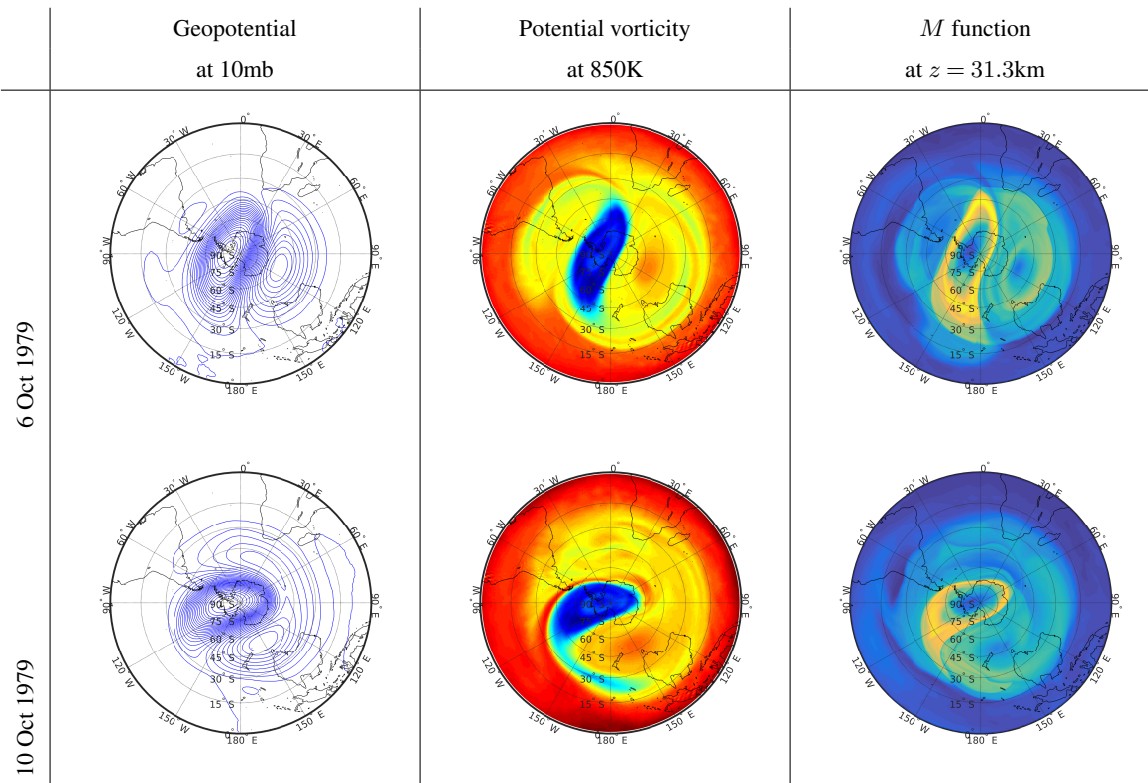

**Figure 4.** Comparison of Geopotential at constant Pressure 10mb, Potential Vorticity at 850K and $M$ function at $z = 31.3$km for several days of October 1979

evolution of the three patches: the one in cyan shows filamentation, the other staying coherent (in blue) and the third done showing a tongue formation (in red) similar to the one observed for the Potential Vorticity in Figure 4. Colors are chosen to highlight the analogous features visible in Figure 4. Clearly Figures 5 b) and c) show time dependent patterns, however the Lagrangian skeleton shown in a) is stationary.

## 5  3D Lagrangian structures over Antarctica

The strong and cyclonic SPV characteristic of the winter circulation above Antarctica has been typically represented in the literature by cross sections such as those in Figs. 2 and 4. In this section we will improve this representation with figures that are more revealing of the full 3D description of the circulation. Figure 6 shows for a day in late winter 1979 (15th of August) the representation of $M$ obtained for $\tau = 5$ for the vertical slice passing through longitudes $90^o$W and $90^o$E. An outstanding feature in this representation is the bright yellow color highlighting a coherent structure. This feature captures the SPV as a tubular structure, similar to those described in Figure 1b), with walls around $60^0$S and an approximately vertical axis coinciding with that of the Earth and extending from the uppermost level of data down to between 15 and 20 km height, i.e. the transition

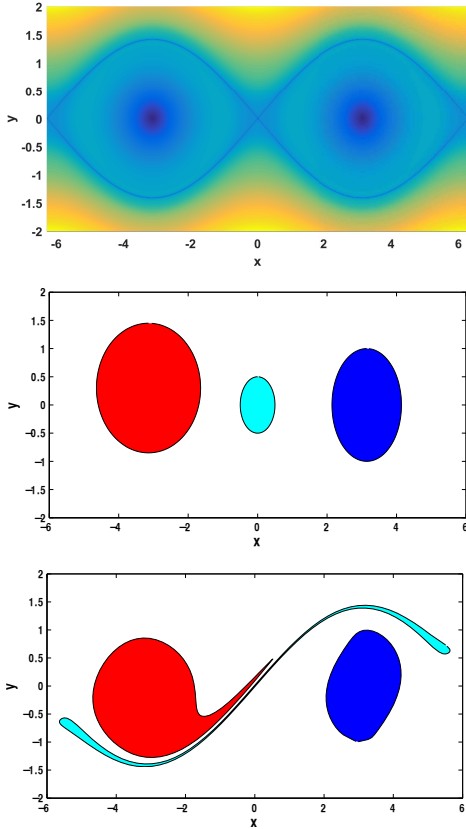

**Figure 5.** a) Lagrangian skeleton for the stationary cat eyes; b) three initial patches of a purely advective field; c) evolution of the patches at time $t = 5.6$

between the troposphere and stratosphere (tropopause). In this case however the average of $M$ does not converge as the flow is aperiodic (see Lopesino *et al.* (2017)), and thus contours of $M$ are not strictly representing invariant sets. Despite this, the setting is analogous to that described for the vortices in the example of section 2. Additionally, the greenish colors that extend equatorward between 15 and 20 km, both at the west and the east, capture the upper tropospheric subtropical westerly jets.

5 Specifically at the west , the greenish colors extend downwards up to the equator ($90^{o}$W) suggesting that the structures involve the entire atmospheric layer. We can also clearly see evidence of the very different dynamical character of the troposphere and stratosphere. Whilst the former region is practically dominated by the SPV, the latter shows much more fine detail reflected by an intricate line pattern. This tangled pattern is the manifestation of crossings of stable and unstable manifolds, which are associated with strong and fast mixing processes at the lowest atmospheric levels. This image is consistent and complementary

10 to the projection at 10 km presented in Figure 2c) and d) and also to the one described next.

The description of the vortex on the 15th August 1979 is supplemented by Fig. 7, which shows the function $M$ computed for $\tau = 10$ along a vertical slice through latitude $60°$S, where most of the SPV walls are. The intricate structures in the troposphere

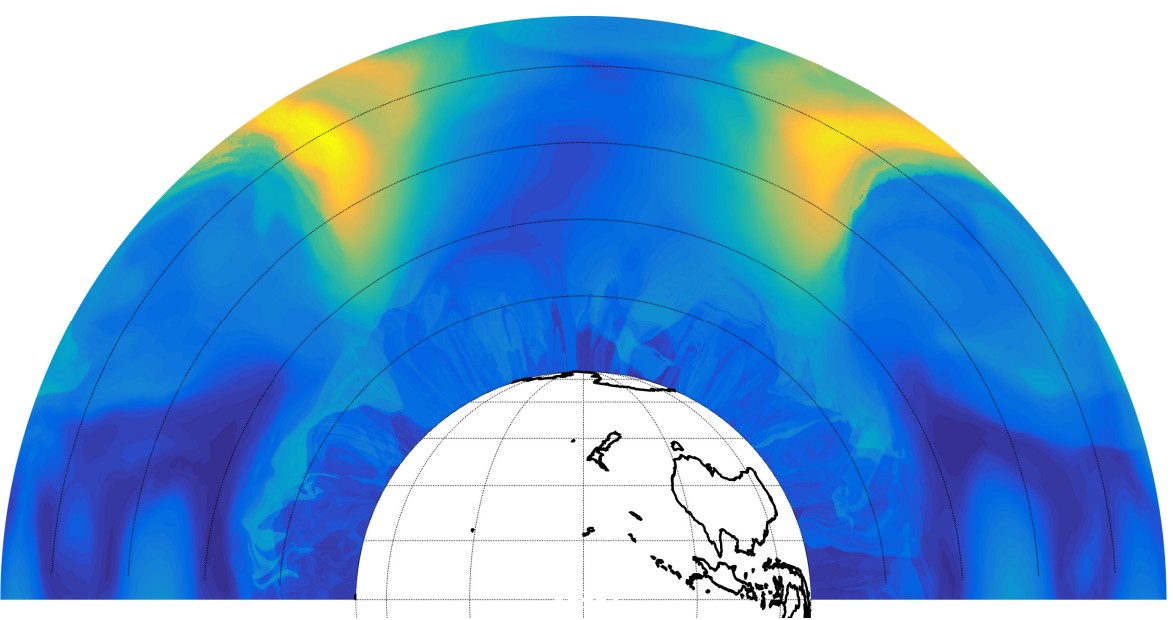

**Figure 6.** 15 August 1979 00:00:00 UTC. $M$ for $\tau = 5$ days displayed along the vertical slice passing through longitudes $90^o$W and $90^o$E. The black lines on the stratosphere correspond to heights 10, 20, 30 and 40km and black lines on the Earth's surface correspond to latitudes $15^o, 30^o, 45^o, 60^o, 75^o$ south.

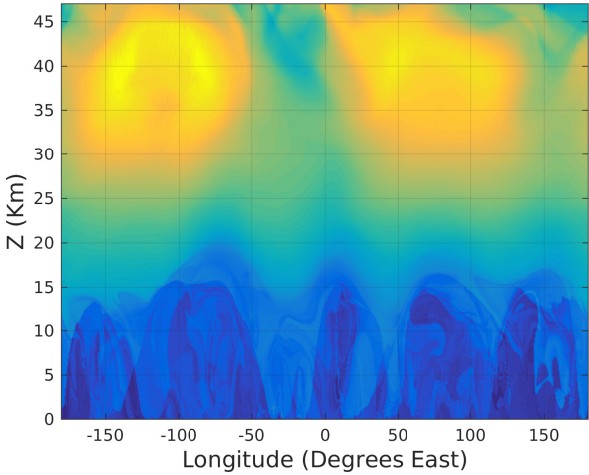

**Figure 7.** 15 August 1979 00:00:00 UTC. $M$ for $\tau = 5$ days displayed along the vertical slice passing through latitude $60^\circ$S.

are also apparent in this figure. In addition, a wavy structure is clearly visible at the boundary between troposphere and stratosphere. In terms of a Fourier decomposition of $M$ at constant height in Fig. 7, we can see the classical pattern of longer wavelengths dominating the field as height increases. At tropopause level, a wavenumber 4 component is clearly visible while at the upper part of the vortex a wavenumber 2 is evident (Manney *et al.*, 1991). The vertical propagation of these features

across the stratosphere from the 8th to the 21st of August of 1979 is clearly visible in the attached movie S1.

We next focus on the description of the 3D Lagrangian structures for the period 6-18 October 1979. Figure 8 shows the rapid changes of the SPV that took place in October 1979 as the lower polar stratosphere warmed up strongly during the spring season (Yamazaki and Mechoso, 1985). The figure shows the function $M$ computed for $\tau = 5$ along a vertical slice passing through latitude 60°S for several October dates. Subplots a), b), c) and d) in Figure 8 confirm that Lagrangian structures in the

stratosphere become more complex in the warming period. On the 18th of October no yellow coherent features are visible in the upper stratosphere.

To help in the interpretation of Fig. 8, Fig. 9 displays horizontal sections of $M$ on the 6th of October at different heights $z = 10$km, $21.2$km, $31.3$km and $40$km as well as a vertical section along meridians $90^o$E, $90^o$W. Figures 8 and 9 show important differences with the winter conditions two months earlier visible in Figs. 6 and 7. The 60°S section in Fig. 8 a) intersects

the cyclonic vortex four times mostly in the western hemisphere. Another deep, anticyclonic vortex appears in the eastern hemisphere above 25km. These cyclonic and anticyclonic vortices are also evident in Fig. 9a) at $z = 31.3$km and $z = 40$km, and in Fig. 9b). In particular in the projection of Fig. 9a) at $z = 31.3$km, along the longitude $0^o$, between latitudes $30^o$- $45^o$ south, crossing lines marked with a white arrow show the presence of a hyperbolic point, being its unstable manifold the separatrix between the two vortices. The vertical extension of this hyperbolic trajectory is depicted with the red line in Figure

9c). This figure shows the analogue to the normally hyperbolic invariant curve explained in Section 2 with the additional feature that here the flow is time dependent, while, for simplicity, the example in Section 2 was stationary. Figure 1b) sketches the correspondant typical configuration for two counter-rotating vortex tubes illustrating this description.

The vertically extended unstable manifold of the normally hyperbolic invariant curve that separates the two vortex tubes is captured by $M$ and is visible in Figure 8 a) and Fig. 9b) as a narrow dark blue line in an analogous way to that presented in

Figure 1. The first of these two figures shows it near the edge of the polar vortex, around $45°$E marked with a white arrow, and for the latter also a white arrow points to the described feature. This manifold structure is separating the two counterrotating vortex tubes just described. It acts as a vertical barrier, which is several kilometers deep as is the case for the unstable manifolds associated to a normally hyperbolic invariant curve formed of hyperbolic trajectories at each $h$ level. Further information on the singular features is given in Fig. 10. The black dots in Figs. 10a) and b) represents for the 6th of October 1979, 00:00:00 UTC,

the horizontal and vertical positions of a particle located on the feature indicated by the blue dark line at a height of 31.3km, longitude $90°$E and latitude $77.74°$S. The black dots in Figs. 10c) and d) show the corresponding locations of the same particle six hours later. The invariant character of the singular structure is confirmed as the particle remains on it during its evolution, and its unstable character is confirmed by the fact that the particle moves away from the hyperbolic point.

Mechoso and Hartmann (1982) have suggested that the fact that the preferred geographical location (ridge south of Australia)

for the development of this anticyclonic vortex in this particular event indicates that the stratospheric circulation is governed to a

significant extent from below (see also Quintanar and Mechoso (1995)). This anticyclonic vortex will strengthen and eventually dominate at high levels. In terms of Fourier components, a quasi-stationary wave 1 amplifies on this date, in conjunction with the displacement of the cyclonic vortex from the polar position.

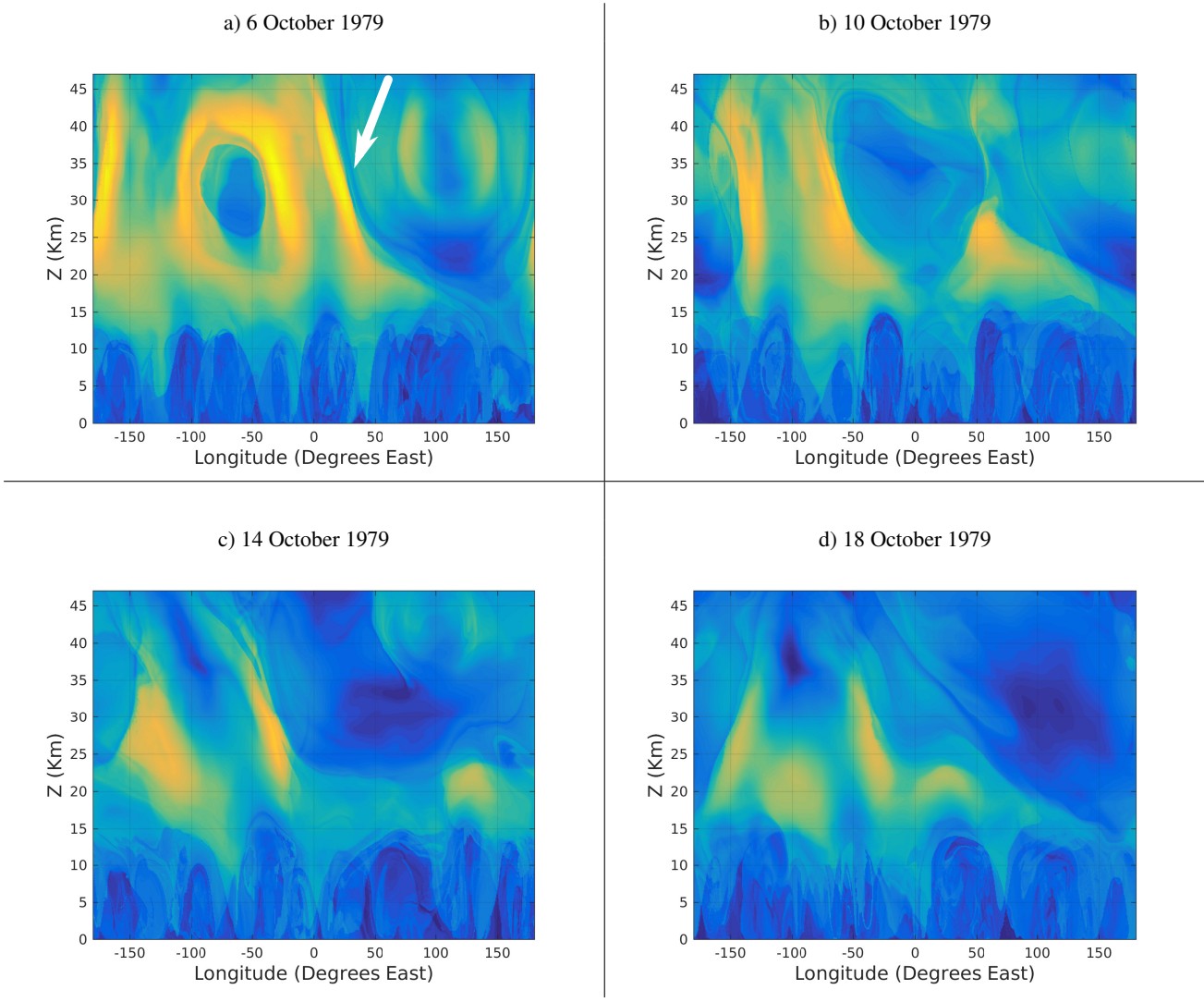

**Figure 8.** Vertical slices showing $M$ for $\tau = 5$ at constant latitude $60°$S at four selected days in October. The color scale is the same in all figures.

# 6 Conclusions

In the present paper we discuss the visualization of three-dimensional Lagrangian structures in atmospheric flows. Specifically, we have explained mathematical aspects about the Lagrangian geometrical structures to be expected in the atmospheric setting in 3D and have introduced the concept of normally hyperbolic invariant curves in an specific example which recover features of those observed in the stratosphere. The algorithm used to represent the 3D Lagrangian structures is based on the methodology of Lagrangian descriptors (LDs). We have explored the application of the full power of the function $M$ computed with 3D trajectories, which hitherto had been used in 2D settings. The consistency of our development has been verified by comparing the 3D scenario results at a constant height with those obtained from the 2D scenario in potential temperature surfaces at equivalent heights.

To demonstrate the methodology we have applied it to a numerical dataset describing the flow above Antarctica during the southern mid-late winter and spring. The dataset was obtained from ERA-Interim Reanalysis data provided by the ECMWF. Our findings show the vertical extension and structure of the stratospheric polar vortex and its evolution. We also characterize, from the Lagrangian point of view, the boundary between the troposhere and the stratosphere. Very complex Lagrangian patterns are identified in the troposphere, which support the presence of strong mixing processes. The "final stratospheric warming" is characterized by the breakdown of the westerly SPV during the transition from winter to summer circulation. Our results confirm that the onset of this process is characterized by an initial decay of the vortex in the upper stratosphere where the circulation weakens, albeit it remains strong at lower heights. We have also captured the anticyclonic circulation that develops during October preferentially above the southern part of Australia. We illustrate the vertical structure of these two counterrotating vortices, and the invariant separatrix that divides them. The particular feature found is several kilometers deep and we demonstrated that fluid parcels remain in this feature during intervals in the order of days. Such features highlight the complexities in the transport of chemical tracers in the stratosphere.

*Acknowledgements.* J. Curbelo, V. J. García-Garrido and A. M. Mancho are supported by MINECO grant MTM2014-56392-R. C. Niang acknowledges Fundacion Mujeres por Africa and ICMAT Severo Ochoa project SEV-2011-0087 for financial support. A. M. Mancho and C. Niang are supported by CSIC grant COOPB20265. The research of S. Wiggins is supported by ONR grant No. N00014-01-1-0769. C. R. Mechoso was supported by the U.S. NSF grant AGS-1245069. We also acknowledge support from ONR grant No. N00014-16-1-2492.

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

a)

z = 10km     z = 21.2km     z = 31.3km     z = 40km

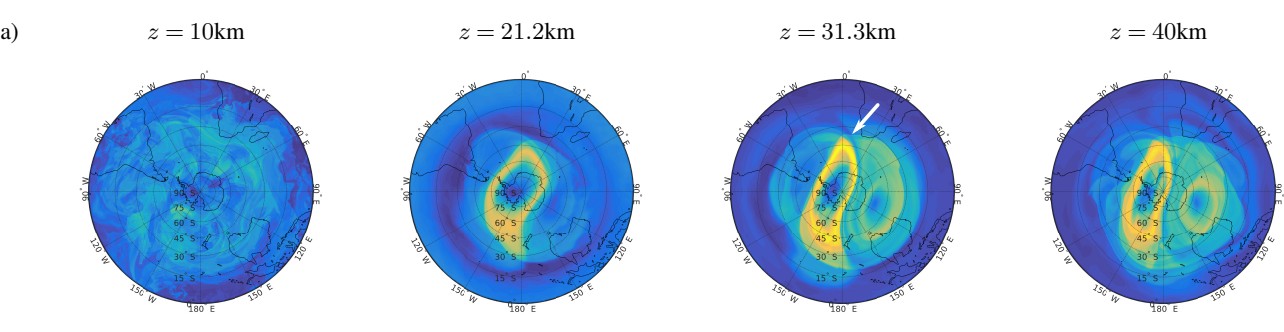

b)

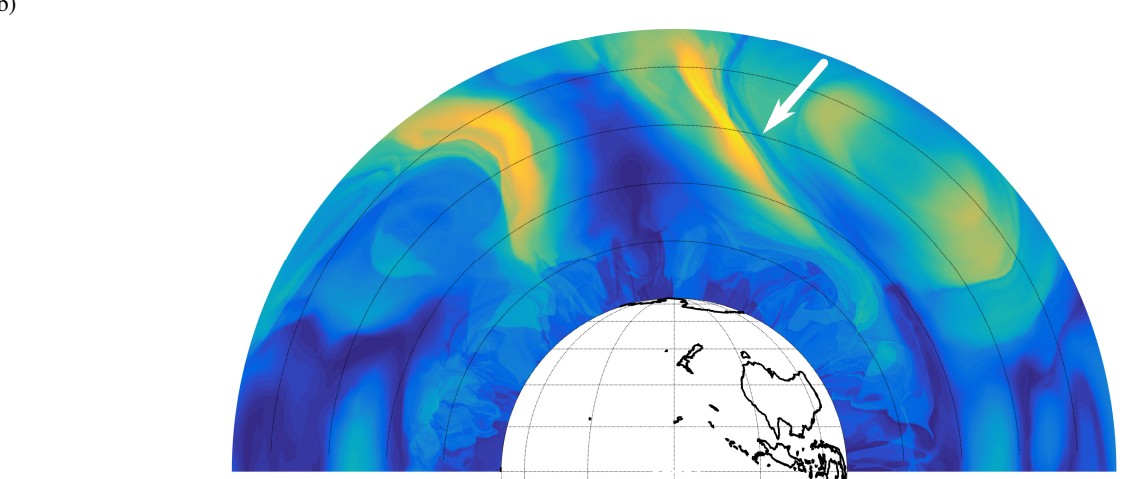

c)

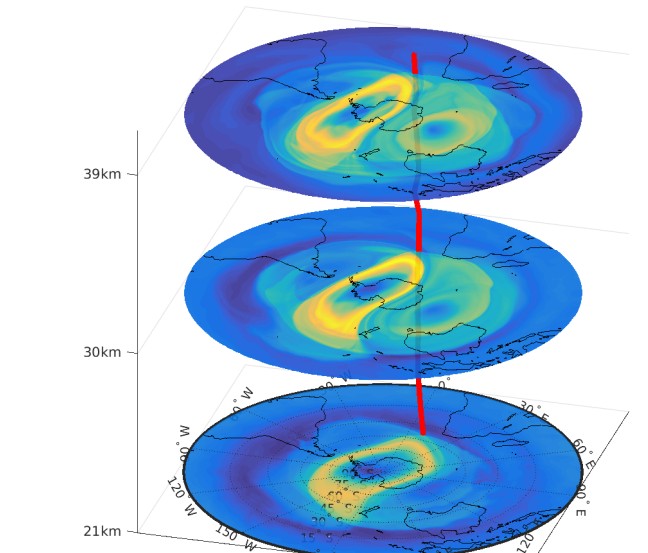

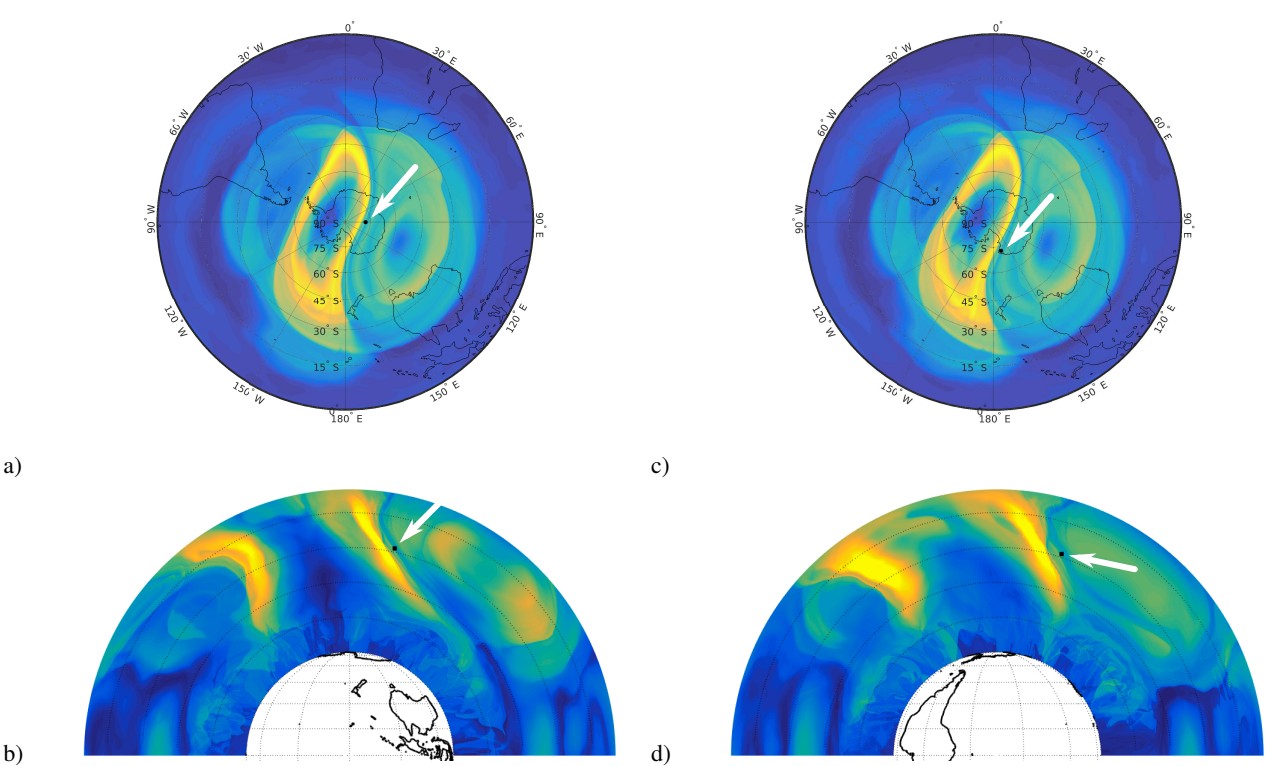

**Figure 10.** Evaluation of the $M$ function with a black particle on it. a) The black particle is placed exactly over an invariant manifold on a 2D slice obtained at height of 31.3 km on the 6 of October 1979 00:00:00 UTC; b) the same black particle on the same day and time placed on a 2D slice obtained at longitude $90^o E$; c) the same black particle six hours later on a 2D slice of $M$ obtained at the corresponding height of the particle at that time; d) the same black particle six hours later on a 2D slice of $M$ obtained at the corresponding longitude of the particle at that time.