# Peer review of "Insights on the three-dimensional Lagrangian geometry of the Antarctic Polar Vortex"

_Nonlinear Processes in Geophysics, 2017_

## Referee Comment (RC1) · Anonymous Referee #1 · 7 Mar 2017

The paper presents an algorithm for the integration of air parcel trajectories in three dimensions and the computation of the "function M' for the analysis of Lagragian transport, and it is applied to atmospheric reanalyzed data for the study of the Antarctic stratospheric vortex. The paper is clearly written, and visualization of the Lagrangian geometry of the stratospheric vortex shows potential. However, the paper lacks of new scientific results. While I cannot recommend publication, I encourage the authors to address my concerns below and resubmit.

The paper consists of two parts. The first one (sections 2, 3 and 4.1), the data processing and methods for computing the Lagrangian geometry are presented. In the second part, the 3D function M is applied to the study of transport in the Antarctic polar vortex.

Regarding the first part, the authors state in the abstract "The present paper introduces

an algorithm for the visualization, analysis and verification of transport and mixing processes in three-dimensional atmospheric flows". While sections 2 and 3 present the methodology in a very clear and concise way, I do not a find significant novelty in this part of the paper.

- Neither the reanalysis data processing, nor the parcel trajectory methods are new (the way they handle the singularity at the poles in geographical coordinates seems identical to that published by some of the co-authors in De la Camara et al. 2012).

- The authors have published multiple articles analyzing transport in oceanic and atmospheric flows using the function M to highlight the Lagrangian geometry of such flows. The extension to 3D, while interesting, does not constitute a new advance from a methodological point of view since the authors have already introduced it in at least a couple of studies (Mancho et al. 2013, Lopesino et al. 2017). Besides, the function M is conceptually defined for n-dimensional fields (Mancho et al. 2013).

- I do not quite understand the verification part of the study. Section 4.1 visually compares maps of the function M obtained from 2D isentropic calculations and from full 3D calculations (Fig 1). But, if I am not mistaken, the authors use the same set of equations (5) for both the 2D and 3D calculations; the only difference is that in 2D the vertical velocity w is zero. Does this mean the authors verify their 3D integration code against itself?

- Figure 2 does a much better job at verifying the function M code by comparing maps of M with geopotential height and potential vorticity (PV) fields. Sadly, the authors do not discuss this figure at all (see lines 16-20). Why is the anticyclone that we see in the height field not visible in the PV or M fields? What are the expected differences between PV and M? What about the tongue of high PV (red

colors) wrapping around the vortex? Why is there no equivalent structure in the M map?

Also, it is rather confusing to analyze maps of these three diagnostics, each using different vertical coordinates. I recommend the authors to interpolate the data to a common horizontal surface.

Regarding the second part (section 4.2), where transport in the Antarctic polar vortex is studied, I have a couple of major concerns the authors might address.

1) The added value of the 3D calculations for the study of the stratospheric transport is not properly addressed. More specifically:

- It is argued throughout the paper that stratospheric motions are basically isentropic for timescales of 10 days. While I understand the authors choice of an integration time of 5 days in Fig. 1 to compare against isentropic trajectories, why using again tau = 5 days in section 4.2 to analyze the 3D Lagrangian geometry of the vortex? Are the results not practically identical to 2D isentropic calculations if the trajectories are integrated over time periods when the isentropic assumption is valid? A way of checking this point would be to perform isentropic calculations at different vertical levels in the stratosphere, and plot similar figures 3-8 (longitude versus height).

- I strongly encourage the authors to increase tau to explore the full potential of 3D calculations. Is there a value of tau over which the Lagrangian geometry from 2D (isentropic) and 3D significantly and 3D diverge?

- Besides, one problem of using the wind field in geometrical height coordinates is that it is difficult to assess what part of the vertical motion is due to the vertical displacement of isentropic surfaces. To prove the added value of the 3D calculations, I recommend using the velocity field in potential temperature surfaces (the vertical velocity would therefore be the heating rate, see Diallo et al 2012 ACP).

2) I do not see any new insights into the dynamics or transport of the polar vortex.

- The description of the evolution of the stratospheric flow during the spring season is, as the authors acknowledge, basically the same as that given in much earlier studies. What have we learned from the analysis of the function M? Is this description richer from that offered by dynamically relevant fields such as geopotential height or PV? I am afraid not.

- One needs a very trained eye to see the geometrical structures that the authors highlight in Figs. 7 and 8 (hyperbolic trajectories and invariant manifolds). How are these structures identified? How is its hyperbolic nature assigned? Also, those structures seem to be located in the outer side of the westerly jet, just in the region where Joseph and Legras (2002 JAS), with similar tools, described a region of chaotic motions. I think this paper does not offer new significant insights into the nature of this region. The fact that the authors identify vertical transport barriers (hardly seen by untrained eyes) does not mean that the motions responsible for those structures are three-dimensional. Again, a detailed comparison between 3D and isentropic calculations is needed.

---

## Referee Comment (RC2) · Anonymous Referee #2 · 15 Mar 2017

**Summary**

In "Insights on the three-dimensional Lagrangian geometry of the Antarctic Polar Vortex" the authors analyze the full 3D Stratospheric Polar Vortex as it weakened during the southern spring of 1979. The analysis was performed using a the M function. The M function is defined as

$$M(\mathbf{x}_0, t_0, \tau) = \int_{t_0-\tau}^{t_0+\tau} ||\mathbf{v}(\mathbf{x}(t; \mathbf{x}_0), t)|| dt,$$

where $\mathbf{v}$ is the velocity vector and $|| \cdot ||$ gives the magnitude of the vector. The M function gives the arc length of the trajectory of the point $\mathbf{x}_0 = \mathbf{x}(t_0)$ over the time interval $(t_0 - \tau, t_0 + \tau)$. High values of M correspond to large displacements in

fluid particles over the integration time, while low values of M correspond to small displacements. Sharp changes in the M field are indicative of stable and unstable manifolds. The velocity field for this study came from the ERA-Interim Reanalysis dataset published by the European Center for Medium-Range Weather Forecasts. This study is notable for its use of a fully 3D velocity field. Previous studies using the M function had mostly focused on 2D flows.

The authors found that the M function was able to accurately detect the Stratospheric Polar Vortex in the full 3D velocity field. The M function was also able to show the weakening of the polar vortex over the observed period. Furthermore, using the M function the authors were able to identify lobes surrounding the polar vortex. The M function was also able to depict the qualitative differences between the stratosphere, dominated by the polar vortex, and the troposphere, dominated by more turbulent and hyperbolic features. The authors conclude that this method can offer accurate insights into the behavior of 3D fluid flows.

**Issues**

The claim is that structures are identified in a 3D flow. But I would expect to see some extracted 2D structures, as was done in the cited paper du Toit, P. C. and Marsden, J. E. (2010). However, this is not the case. We merely see cross-sections of what are presumably 2D structures in the 3D flow.

How to identify elliptical LCS from the M function should be stated more clearly.

The use of the term algorithm in this paper is a bit confusing. Usually one expects to see a set of step by step instructions or a flow chart associated with an

algorithm.

**Compliments**

This paper was very well written.

The naming of the software tools and commands that were used in this was very helpful.

Pointing out the effect that terrain can have on stratospheric phenomena was a nice touch.

---

## Author Response (AR1)

**Answer to Referee 1**

We wish to thank to this referee for his/her very useful comments, which have helped us to improve the manuscript, and have been addressed as follows:

**General comments:**

**1.** *The paper presents an algorithm for the integration of air parcel trajectories in three dimensions and the computation of the function M for the analysis of Lagrangian transport, and it is applied to atmospheric reanalyzed data for the study of the Antarctic stratospheric vortex. The paper is clearly written, and visualization of the Lagrangian geometry of the stratospheric vortex shows potential. However, the paper lacks of new scientific results. While I cannot recommend publication, I encourage the authors to address my concerns below and resubmit.*

We have clarified in the new version of the article the major goal, as maybe this was not sufficiently clear in the original manuscript. In the new version we state very clearly that the goal is to describe 3D Lagrangian structures on the stratosphere. To this end we have described in more detail (Section 2) what kind of 3D Lagrangian structures are expected in the stratosphere and we list specifically what are the new scientific results in this regard in the Abstract, Section 5 and the Conclusions.

**2.** *Regarding the first part, the authors state in the abstract "The present paper introduces an algorithm for the visualization, analysis and verification of transport and mixing processes in three-dimensional atmospheric flows". While sections 2 and 3 present the methodology in a very clear and concise way, I do not a find significant novelty in this part of the paper.*

• *Neither the reanalysis data processing, nor the parcel trajectory methods are new (the way they handle the singularity at the poles in geographical coordinates seems identical to that published by some of the co-authors in De la Camara et al. 2012).*

• *The authors have published multiple articles analyzing transport in oceanic and atmospheric flows using the function M to highlight the Lagrangian geometry of such flows. The extension to 3D, while interesting, does not constitute a new advance from a methodological point of view since the authors have already introduced it in at least a couple of studies (Mancho et al. 2013, Lopesino et al. 2017). Besides, the function M is conceptually defined for n-dimensional fields (Mancho et al. 2013).*

The referee is right. The algorithm is not new, and that sentence has been modified. The previous applications of the function $M$ to 3D flows are summarised between lines 19 and 25 in the Introduction. Regarding the algorithm, what is new is its implementation for analysing 3D atmospheric data sets as stated in line 36 of the Introduction. We mantain the section describing the implementation of the algorithm (Section 3) because it may be useful to other researchers interested in applying this tool for similar purposes.

**3.** *I do not quite understand the verification part of the study. Section 4.1 visually compares maps of the function M obtained from 2D isentropic calculations and from full 3D calculations (Fig 1). But, if I am not mistaken, the authors use the same set of equations (5) for both the 2D and 3D calculations; the only difference is that in 2D the*

*vertical velocity w is zero. Does this mean the authors verify their 3D integration code against itself?*

The 2D calculation is done on constant potential temperature surfaces which in general are surfaces which are time dependent and do not coincide with spherical shells. The velocity fields on these surfaces are also downloaded from ERA-interim. The 3D calculation is done in the 3D space with 3D velocity fields downloaded from ERA-interim and processed as explained in the article. Section 4 now discusses these issues and has been extended to include calculations showing cases in which the 3D calculation on spherical shells and the 2D calculation on the constant potential temperature surfaces coincide (upper stratosphere) and where they do not (upper troposphere).

**4.** *Figure 2 does a much better job at verifying the function M code by comparing maps of M with geopotential height and potential vorticity (PV) fields. Sadly, the authors do not discuss this figure at all (see lines 16-20). Why is the anticyclone that we see in the height field not visible in the PV or M fields? What are the expected differences between PV and M? What about the tongue of high PV (red colors) wrapping around the vortex? Why is there no equivalent structure in the M map?*

*Also, it is rather confusing to analyze maps of these three diagnostics, each using different vertical coordinates. I recommend the authors to interpolate the data to a common horizontal surface.*

Figure 2 (now figure 3) has been thoroughly explained in Section 4. The issues rised by the referee have been addressed including a new figure.

The different vertical coordinates used for each map are standard in atmospheric sciences and we are plotting for each one the value which is in correspondence to their partners. We do not think that differences in the units is a problem.

**5.** *It is argued throughout the paper that stratospheric motions are basically isentropic for timescales of 10 days. While I understand the authors choice of an integration time of 5 days in Fig. 1 to compare against isentropic trajectories, why using again tau = 5 days in section 4.2 to analyze the 3D Lagrangian geometry of the vortex? Are the results not practically identical to 2D isentropic calculations if the trajectories are integrated over time periods when the isentropic assumption is valid? A way of checking this point would be to perform isentropic calculations at different vertical levels in the stratosphere, and plot similar figures 3-8 (longitude versus height).*

We have compared isentropic calculations and 3D calculations in the new Figures 2 and 3 in Section 4 and it is shown when they coincide or not. We also discuss the effect of the integration period $\tau$ on $M$. Additionally we have explained more clearly what new insights are brought by our results. Indeed, the 3D calculations allow to perform slices in directions perpendicular to plane of motion and those sections highlight the structure of normally hyperbolic invariant objects whose vertical structure cannot be visualized otherwise. To our knowledge such visualisation is described for the first time in this article.

**6.** • *I strongly encourage the authors to increase tau to explore the full potential of 3D calculations. Is there a value of tau over which the Lagrangian geometry from 2D (isentropic) and 3D significantly and 3D diverge?*

We have done this in Section 4.

**7.** *Besides, one problem of using the wind field in geometrical height coordinates is that it is difficult to assess what part of the vertical motion is due to the vertical displacement of isentropic surfaces. To prove the added value of the 3D calculations, I recommend using the velocity field in potential temperature surfaces (the vertical velocity would therefore be the heating rate, see Diallo et al 2012 ACP).*

The wind field in geometrical coordinates is useful to integrate equations (7). The integration of this system has allowed us to describe 3D Lagrangian structures in the stratosphere. In particular we describe to our knowledge for the first time the following issues: vertical extension of the stratospheric polar vortex and its lower limit and its tilted character. We have identified the boundary between troposphere and stratosphere. We have identified lagrangian structures, fully 3D, showing strong mixing into the troposphere. We have discussed the vertical structures of two counterrotating vortices, (the polar vortex and a new emerging one) and identified an invariant structure separating them and have related this to the presence of a normally hyperbolic invariant curve. For all this purposes our approach has shown to be sufficient and consistent with other results, thus we do not think it is necessary to repeat calculations within another approach. To address the issues regarding transport across potential temperature surfaces is a very interesting question feasible also within our approach (just calculating $M$ on the time dependent potential temperature surfaces) but out of the scope of the current manuscript.

**8.** *2) I do not see any new insights into the dynamics or transport of the polar vortex.*

• *The description of the evolution of the stratospheric flow during the spring season is, as the authors acknowledge, basically the same as that given in much earlier studies. What have we learned from the analysis of the function M? Is this description richer from that offered by dynamically relevant fields such as geopotential height or PV? I am afraid not*

We have explained in the current version very clearly what are the novel insights on the stratosphere provided by our work. It might be that our discussions in the first version were too much focused in addressing consistency with previous findings, and our findings were not sufficiently emphasized. For this reason we have rewritten the manuscript to address these issues. A list of specific new insights are described in the 7th bullet point, and further discussion about the comparison with the geopotential height and PV is given in the new version.

**9.** *One needs a very trained eye to see the geometrical structures that the authors highlight in Figs. 7 and 8 (hyperbolic trajectories and invariant manifolds). How are these structures identified? How is its hyperbolic nature assigned? Also, those structures seem to be located in the outer side of the westerly jet, just in the region where Joseph and Legras (2002 JAS), with similar tools, described a region of chaotic motions. I think this paper does not offer new significant insights into the nature of this region. The fact that the authors identify vertical transport barriers (hardly seen by untrained eyes) does not mean that the motions responsible for those structures are three-dimensional. Again, a detailed comparison between 3D and isentropic calculations is needed.*

We have introduced a new Section 2 which introduces and mathematically describes the type of 3D Lagrangian structures expected in the stratosphere. A relevant example is introduced. We have added extra arrows in current figures 8 and 9 to highlight the

geometrical structures we are interested in and those are linked with the example described in the new Section 2.

Figure 9 a) is similar to the projections performed by Josep and Legras, although in that article they do not discuss the event we address in this figure about the boundary separating two counterrotating vortices present in the atmosphere. The advantage of the used tools with respect to the FSLE used by Joseph and Legras is that the $M$ function highlights simultaneously manifolds and coherent structures related to elliptic regions.

On the other hand figures showing the vertical structure across the stratosphere are new and to our knowledge have not been described before.

Figure 8 address the invariant character of structures related to sharp changes in the color code of $M$. This is done by tracking a particle trajectory and observing that during its evolution, its position stays on a line with an abrupt change in the $M$ color code.

A detailed comparison between 3D and isentropic calculations is addressed now in figures 3 and 4.

**Answer to Referee 2**

We wish to thank to this referee for his/her very useful comments, which have helped us to improve the manuscript, and have been addressed as follows:

**Issues:**

**1.** *The claim is that structures are identified in a 3D flow. But I would expect to see some extracted 2D structures, as was done in the cited paper du Toit, P. C. and Marsden, J. E. (2010). However, this is not the case. We merely see cross-sections of what are presumably 2D structures in the 3D flow.*

The Introduction of the new version of the manuscript explains (from lines 8 to 28) different approaches used to identify 3D Lagrangian structures in 3D flows. The one by du Toit and Marsden is one but not the only one. In the new Section 2 these issues raised by the referee are carefully addressed. Our approach consist of the use of function $M$ and this methodology gains insights in the 3D flow by computing the function $M$ on slices with different orientation. This approach does not compute surfaces representing the 2D invariant manifolds, but obtains them by means of slices and has the advantage of highlighting tori-like structures.

**2.** *How to identify elliptical LCS from the M function should be stated more clearly.*

We have done this in Section 2.

**3.** *The use of the term algorithm in this paper is a bit confusing. Usually one expects to see a set of step by step instructions or a flow chart associated with an algorithm.*

We have done this in Section 3.2.

[revised manuscript text omitted]

Aurell, E., Boffeta, G., Crisanti, A., Paladin, G., and Vulpiani, A. (1997). Predictability in the large: An extension of the concept of Lyapunov exponent. *J. Phys. A :Math. Gen.*, **30**, 1–26.

Bettencourt, J. H., López, C., Hernández-García, E., Montes, I., Sudre, J., Dewitte, B., Paulmier, A., and Garçon, V. (2014). Boundaries of the peruvian oxygen minimum zone shaped by coherent mesoscale dynamics. *Nature Geoscience*, **8**, 937–940.

Bower, A. S. (1991). A simple kinematic mechanism for mixing fluid parcels across a menadering jet. *J. Phys. Oceanogr.*, **21**, 173–180.

Bowman, K. P. (1993). Large-scale isentropic mixing properties of the Antarctic polar vortex from analyzed winds. *Journal of Geophysical Research*, **98**, 23013–23027.

Bowman, K. P. (2006). Transport of carbon monoxide from the tropics to the extratropics. *Journal of Geophysical Research: Atmospheres*, **111**(D2).

Branicki, M. and Kirwan Jr., A. D. (2010). Stirring: The Eckart paradigm revisited. *Int. J. Eng. Sci.*, **48**, 1027–1042.

Branicki, M. and Wiggins, S. (2009). An adaptive method for computing invariant manifolds in non-autonomous, three-dimensional dynamical systems. *Physica D*, **238**(16), 1625 – 1657.

Branicki, M., Mancho, A. M., and Wiggins, S. (2011). A Lagrangian description of transport associated with a front-eddy interaction: application to data from the North-Western Mediterranean sea. *Physica D*, **240**(3), 282–304.

Cartwright, J. H. E., Feingold, M., and Piro, O. (1996). Chaotic adection in three-dimensional unsteady incompressible laminar flow. *J. Fluid Mech.*, **316**, 259–284.

Charlton, A. J., O'Neill, A., Lahoz, W. A., and Berrisford, P. (2006). The splitting of the stratospheric polar vortex in the southern hemisphere, september 2002: Dynamical evolution. *J. Atmos. Sci.*, **66**, 590–602.

de la Cámara, A., Mancho, A. M., Ide, K., Serrano, E., and Mechoso, C. (2012). Routes of transport across the Antarctic polar vortex in the southern spring. *J. Atmos. Sci.*, **69**(2), 753–767.

de la Cámara, A., Mechoso, R., Mancho, A. M., Serrano, E., and Ide., K. (2013). Isentropic transport within the Antarctic polar night vortex: Rossby wave breaking evidence and Lagrangian structures. *J. Atmos. Sci.*, **70**, 2982–3001.

Dee, D. P. *et al.* (2011). The ERA-Interim reanalysis: configuration and performance of the data assimilation system. *Quarterly Journal of the Royal Meteorological Society*, **137**(656), 553–597.

d'Ovidio, F., Isern-Fontanet, J., López, C., Hernández-García, E., and García-Ladona, E. (2009). Comparison between Eulerian diagnostics and finite-size Lyapunov exponents computed from altimetry in the Algerian basin. *Deep Sea Res. I*, **56**(1), 15–31.

Dritschel, D. G. (1989). Contour dynamics and contour surgery: numerical algorithms for extended, high-resolution modelling of vortex dynamics in two-dimensional, inviscid, incompressible flows. *Comput. Phys. Rep.*, **10**, 77–146.

du Toit, P. C. and Marsden, J. E. (2010). Horseshoes in hurricanes. *J. Fixed Point Theory Appl.*, **7**, 351–384.

Farazmand, M. and Haller, G. (2012). Computing Lagrangian Coherent Structures from variational LCS theory. *Chaos*, **22**, 013128.

Garcia-Garrido, V. J., Mancho, A. M., and Wiggins, S. (2015). A dynamical systems approach to the surface search for debris associated with the disappearance of flight MH370. *Nonlin. Proc. Geophys.*, **22**, 701–712.

Garcia-Garrido, V. J., Ramos, A., Mancho, A. M., Coca, J., and Wiggins, S. (2016). A dynamical systems perspective for a real-time response to a marine oil spill. *Marine Pollution Bulletin.*, pages 1–10.

García-Garrido, V. J., Curbelo, J., Mechoso, C. R., Mancho, A. M., and Wiggins, S. (2017). A simple kinematic model for the lagrangian description of relevant nonlinear processes in the stratospheric polar vortex. *Nonlin. Proc. Geophys. Discussion*.

Guha, A., Mechoso, C. R., Konor, C. S., and Heikes, R. P. (2016). Modeling rossby wave breaking in the southern spring stratosphere. *J. Atmos. Sci.*, **73**(1), 393–406.

Haller, G. (2000). Finding finite-time invariant manifolds in two-dimensional velocity fields. *Chaos*, **10**(1), 99–108.

Haller, G. (2001). Distinguished material surfaces and coherent structures in three-dimensional fluid flows. *Physica D*, **149**, 248–277.

Haller, G. and Beron-Vera, F. J. (2012). Geodesic theory of transport barriers in two-dimensional flows. *Physica D*, **241**(7), 1680–1702.

Haller, G. and Yuan, G. (2000). Lagrangian coherent structures and mixing in two-dimensional turbulence. *Physica D*, **147**, 352–370.

Holton, J. R. (2004). *An Introduction to Dynamic Meteorology*. Elsevier Academic Press.

Ide, K., Small, D., and Wiggins, S. (2002). Distinguished hyperbolic trajectories in time dependent fluid flows: analytical and computational approach for velocity fields defined as data sets. *Nonlin. Proc. Geophys.*, **9**, 237–263.

Joseph, B. and Legras, B. (2002). Relation between kinematic boundaries, stirring, and barriers for the Antarctic polar vortex. *J. Atmos. Sci.*, **59**, 1198–1212.

Ju, N., Small, D., and Wiggins, S. (2003). Existence and computation of hyperbolic trajectories of aperiodically time-dependent vector fields and their approximations. *Int. J. Bif. Chaos*, **13**, 1449 –1457.

Juckes, M. N. and McIntyre, M. E. (1987). A high-resolution one-layer model of breaking planetary waves in the stratosphere. *Nature*, **328**, 590–596.

Krüger, K., Naujokat, B., and Labitzke, K. (2005). The unusual midwinter warming in the southern hemisphere stratosphere 2002: A comparison to northern hemisphere phenomena. *J. Atmos. Sci.*, **62**, 603–613.

Lekien, F. and Ross, S. D. (2010). The computation of finite-time lyapunov exponents on unstructured meshes and for non-euclidean manifolds. *Chaos*, **20**, 017505.

Lopesino, C., Balibrea-Iniesta, F., Wiggins, S., and Mancho, A. M. (2015). Lagrangian descriptors for two dimensional, area preserving autonomous and nonautonomous maps. *Communications in Nonlinear Science and Numerical Simulations*, **27**(1-3), 40–51.

Lopesino, C., Balibrea-Iniesta, F., García-Garrido, V. J., Wiggins, S., and Mancho, A. M. (2017). A theoretical framework for lagrangian descriptors. to appear. *International Journal of Bifurcation and Chaos*.

Madrid, J. A. J. and Mancho, A. M. (2009). Distinguished trajectories in time dependent vector fields. *Chaos*, **19**, 013111.

Malhotra, N. and Wiggins, S. (1998). Geometric structures, lobe dynamics, and Lagrangian transport in flows with aperiodic time-dependence, with applications to Rossby wave flow. *J. Nonlinear Science*, **8**, 401–456.

Mancho, A. M., Small, D., Wiggins, S., and Ide, K. (2003). Computation of Stable and Unstable Manifolds of Hyperbolic Trajectories in Two-Dimensional, Aperiodically Time-Dependent Vectors Fields. *Physica D*, **182**, 188–222.

Mancho, A. M., Small, D., and Wiggins, S. (2004). Computation of hyperbolic trajectories and their stable and unstable manifolds for oceanographic flows represented as data sets. *Nonlin. Proc. Geophys.*, **11**, 17–33.

Mancho, A. M., Small, D., and Wiggins, S. (2006a). A comparison of methods for interpolating chaotic flows from discrete velocity data. *Computers and Fluids*, **35**, 416–428.

Mancho, A. M., Hernández-García, E., Small, D., Wiggins, S., and Fernández, V. (2006b). Lagrangian transport through an ocean front in the North-Western Mediterranean Sea. *J. Phys. Oceanogr.*, **38**(6), 1222–1237.

Mancho, A. M., Small, D., and Wiggins, S. (2006c). A tutorial on dynamical systems concepts applied to Lagrangian transport in oceanic flows defined as finite time data sets: Theoretical and computational issues. *Phys. Rep.*, **237**(3-4), 55–124.

Mancho, A. M., Wiggins, S., Curbelo, J., and Mendoza, C. (2013). Lagrangian descriptors: A method for revealing phase space structures of general time dependent dynamical systems. *Communications in Nonlinear Science and Numerical Simulations*, **18**(12), 3530–3557.

Manney, G. L. and Lawrence, Z. D. (2016). The major stratospheric final warming in 2016: Dispersal of vortex air and termination of Arctic chemical ozone loss. *Atmospheric Chemistry and Physics Discussions*, **2016**, 1–40.

5    Manney, G. L., Farrara, J. D., and Mechoso, C. R. (1991). The behavior of wave 2 in the southern hemisphere stratosphere during late winter and early spring. *J. Atmos. Sci.*, **48**, 976–998.

Manney, G. L., Sabutis, J. L., Alley, D. R., Lahoz, W. A., Scaife, A. A., Randall, C. E., Pawson, S., Naujokat, B., and Swinbank, R. (2006). Simulations of dynamics and transport during the september 2002 antarctic major warming. *J. Atmos. Sci.*, **66**(690-707).

McIntyre, M. E. and Palmer, T. N. (1983). Breaking planetary waves in the stratosphere. *Nature*, **305**, 593–600.

10    McIntyre, M. E. and Palmer, T. N. (1984). The surf zone in the stratosphere. *Journal of Atmospheric and Terrestrial Physics*, **46**(9), 825–849.

McIntyre, M. E. and Palmer, T. N. (1985). A note on the general concept of wave breaking for rossby and gravity waves. *Pure and Applied Geophysics*, **123**(6), 964–975.

Mechoso, C. R. and Hartmann, D. L. (1982). An observational study of traveling planetary waves in the southern hemisphere. *Journal of the Atmospheric Sciences*, **39**(9), 1921–1935.

15    Mechoso, C. R., O'Neill, A., Pope, V. D., and Farrara, J. D. (1988a). A study of the stratospheric final warming of 1982 in the southern hemisphere. *Quarterly Journal of the Royal Meteorological Society*, **114**(484), 1365–1384.

Mechoso, C. R., O'Neill, A., Pope, V. D., and Farrara, J. D. (1988b). A study of the stratospheric final warming of 1982 in the Southern Hemisphere. *Quart. J. R. Meteor. Soc.*, **114**, 1365–1384.

Mendoza, C. and Mancho, A. M. (2010). The hidden geometry of ocean flows. *Phys. Rev. Lett.*, **105**(3), 038501.

20    Mendoza, C. and Mancho, A. M. (2012). The Lagrangian description of aperiodic flows: a case study of the Kuroshio Current. *Nonlin. Proc. Geophys.*, **19**(14), 449–472.

Mendoza, C., Mancho, A. M., and Wiggins, S. (2014). Lagrangian descriptors and the assesment of the predictive capacity of oceanic data sets. *Nonlin. Proc. Geophys.*, **21**, 677–689.

Mezić, I. and Wiggins, S. (1994). On the integrability and perturbation of three-dimensional fluid flows with symmetry. *Journal of Nonlinear Science*, **4**(1), 157–194.

25    Mezic, I. and Wiggins, S. (1999). A method for visualization of invariant sets of dynamical systems based on the ergodic partition. *Chaos*, **9**(1), 213–218.

Moharana, N. R., Speetjens, M. F. M., Trieling, R. R., and Clercx, H. J. H. (2013). Three-dimensional lagrangian transport phenomena in unsteady laminar flows driven by a rotating sphere. *Phys. Fluids*, **25**, 093602.

30    Morales-Juberías, R., Sayanagi, K. M., Simon, A. A., Fletcher, L. N., and Cosentino, R. G. (2015). Meandering shallow atmospheric jet as a model of saturn's north-polar hexagon. *The Astrophysical Journal Letters*, **806**, L18 (6pp).

Nakamura, M. and Plumb, R. A. (1994). The effects of flow asymmetry on the direction of rossby wave breaking. *J. Atmos. Sci.*, **51**, 2031–2044.

Ottino, J. M. (1989). *The Kinematics of Mixing: Stretching, Chaos, and Transport*. Cambridge University Press, Cambridge, England.
35    Reprinted 2004.

Polvani, L. M. and Plumb, R. A. (1992). Rossby wave breaking, microbreaking, filamentation, and secondary vortex formation: The dynamics of a perturbed vortex. *J. Atmos. Sci.*, **49**(6), 462–476.

Pouransari, Z., Speetjens, M. F. M., and Clercx, H. J. H. (2010). Formation of coherent structures by fluid inertia in three-dimensional laminar flows. *J. Fluid Mech.*, **654**, 5–34.

Press, W. H., Teukolsky, S. A., Vetterling, W. T., and Flannery, B. P. (1992). *Numerical Recipes in C: The Art of Scientific Computing*. Cambridge University Press, New York, NY, USA.

Quintanar, A. I. and Mechoso, C. R. (1995). Quasi-stationary waves in the southern hemisphere. part i: Observational data. *J. Climate*, **4**, 2659–2672.

Rabier, F. *et al.* (2010). The concordiasi project in antarctica. *Bulletin of the American Meteorological Society*, **91**(1), 69–86.

Rempel, E. L., Chian, A. C.-L., Brandenburg, A., Munuz, P. R., and Shadden, S. C. (2013). Coherent structures and the saturation of a nonlinear dynamo. *Journal of Fluid Mechanics*, **729**, 309–329.

Rutherford, B. and Dangelmayr, G. (2010). A three-dimensional lagrangain hurricane eyewall computation. *Quarterly Journal of the Royal Meteorological Society*, **136**, 1931–1944.

Rutherford, B., Dangelmayr, G., and Montgomery, M. T. (2012). Lagrangian coherent structures in tropical cyclone intensification. *Atmospheric Chemistry and Physics*, **12**, 5483–5507.

Rypina, I. I., Brown, M. G., Beron-Vera, F. J., Kocak, H., Olascoaga, M. J., and Udovydchenkov, I. A. (2007). On the lagrangian dynamics of atmospheric zonal jets and the permeability of the stratospheric polar vortex. *J. Atmos. Sci.*, **64**, 3595–3610.

Rypina, I. I., Pratt, L. J., Wang, P., Özgökmen, T. M., and Mezic, I. (2015). Resonanace phenomena in a time-dependent, three-dimensional model of an idealized eddy. *Chaos*, **25**, 087401.

Samelson, R. and Wiggins, S. (2006). *Lagrangian Transport in Geophysical Jets and Waves: The Dynamical Systems Approach*. Springer-Verlag, New York.

Samelson, R. M. (1992). Fluid exchange across a meandering jet. *J. Phys. Oceanogr.*, **22**(4), 431–440.

Shadden, S. C., Lekien, F., and Marsden, J. E. (2005). Definition and properties of Lagrangian Coherent Structures from finite-time Lyapunov exponents in two-dimensional aperiodic flows. *Physica D*, **212**, 271–304.

Simmons, A., Uppala, S., Dee, D., and S, K. (2007). ERA-Interim: New ECMWF reanalysis products from 1989 onwards. *ECMWF Newsletter*, **110**, 25–35.

Smith, M. L. and McDonald, A. J. (2014). A quantitative measure of polar vortex strength using the function m. *J. Gephys.Res. Atmos.*, **119**, 5966–5985.

Snyder, J. P. (1987). *Map Projections–A Working Manual*. U.S. Geological Survey professional paper. U.S. Government Printing Office.

Stewartson, K. (1977). The evolution of the critical layer of a rossby wave. *Geophysical & Astrophysical Fluid Dynamics*, **9**(1), 185–200.

Warn, T. and Warn, H. (1978). The evolution of a nonlinear critical level. *Studies in Applied Mathematics*, **59**(1), 37–71.

Wiggins, S. (1988). *Global bifurcations and chaos: analytical methods*, volume 73. Springer Verlag.

Wiggins, S. (2010). Coherent structures and chaotic advection in three dimensions. *J. Fluid Mech.*, **654**, 1–4.

Yamazaki, K. and Mechoso, C. R. (1985). Observations of the final warming in the stratosphere of the southern hemisphere during 1979. *J. Atmos. Sci.*, **42**, 1198–1205.

c)to c) the particle approaches a hyperbolic point along the stable manifold, which is visible in the different slices. From frame d) onwards the particle r

[Figure]

**Figure 10.** Evaluation of the $M$ function with a black particle on it. a) The black particle is placed exactly over an invariant manifold on a 2D slice obtained at height of 31.3 km on the 6 of October 1979 00:00:00 UTC; b) the same black particle on the same day and time placed on a 2D slice obtained at longitude $90^o E$; c) the same black particle six hours later on a 2D slice of $M$ obtained at the corresponding height of the particle at that time; d) the same black particle six hours later on a 2D slice of $M$ obtained at the corresponding longitude of the particle at that time.